# piRNA Defense Against Endogenous Retroviruses

**DOI:** 10.3390/v16111756

**Published:** 2024-11-09

**Authors:** Milky Abajorga, Leonid Yurkovetskiy, Jeremy Luban

**Affiliations:** 1Program in Molecular Medicine, University of Massachusetts Chan Medical School, Worcester, MA 01605, USA; 2Morningside Graduate School of Biomedical Sciences, University of Massachusetts Chan Medical School, Worcester, MA 01655, USA; 3Department of Biochemistry and Molecular Biotechnology, University of Massachusetts Chan Medical School, Worcester, MA 01605, USA; 4RNA Therapeutics Institute, University of Massachusetts Chan Medical School, Worcester, MA 01605, USA; 5Li Weibo Institute for Rare Diseases Research, University of Massachusetts Chan Medical School, Worcester, MA 01605, USA; 6Broad Institute of MIT and Harvard, Cambridge, MA 02142, USA; 7Ragon Institute of MGH, MIT, and Harvard, Cambridge, MA 02139, USA; 8Massachusetts Consortium on Pathogen Readiness, Boston, MA 02115, USA

**Keywords:** endogenous retrovirus, piRNA, KoRV-A, gammaretrovirus, PIWI, splicing, transposons germline, transcriptional silencing

## Abstract

Infection by retroviruses and the mobilization of transposable elements cause DNA damage that can be catastrophic for a cell. If the cell survives, the mutations generated by retrotransposition may confer a selective advantage, although, more commonly, the effect of new integrants is neutral or detrimental. If retrotransposition occurs in gametes or in the early embryo, it introduces genetic modifications that can be transmitted to the progeny and may become fixed in the germline of that species. PIWI-interacting RNAs (piRNAs) are single-stranded, 21–35 nucleotide RNAs generated by the PIWI clade of Argonaute proteins that maintain the integrity of the animal germline by silencing transposons. The sequence specific manner by which piRNAs and germline-encoded PIWI proteins repress transposons is reminiscent of CRISPR, which retains memory for invading pathogen sequences. piRNAs are processed preferentially from the unspliced transcripts of piRNA clusters. Via complementary base pairing, mature antisense piRNAs guide the PIWI clade of Argonaute proteins to transposon RNAs for degradation. Moreover, these piRNA-loaded PIWI proteins are imported into the nucleus to modulate the co-transcriptional repression of transposons by initiating histone and DNA methylation. How retroviruses that invade germ cells are first recognized as foreign by the piRNA machinery, as well as how endogenous piRNA clusters targeting the sequences of invasive genetic elements are acquired, is not known. Currently, koalas (*Phascolarctos cinereus)* are going through an epidemic due to the horizontal and vertical transmission of the KoRV-A gammaretrovirus. This provides an unprecedented opportunity to study how an exogenous retrovirus becomes fixed in the genome of its host, and how piRNAs targeting this retrovirus are generated in germ cells of the infected animal. Initial experiments have shown that the unspliced transcript from KoRV-A proviruses in koala testes, but not the spliced KoRV-A transcript, is directly processed into sense-strand piRNAs. The cleavage of unspliced sense-strand transcripts is thought to serve as an initial innate defense until antisense piRNAs are generated and an adaptive KoRV-A-specific genome immune response is established. Further research is expected to determine how the piRNA machinery recognizes a new foreign genetic invader, how it distinguishes between spliced and unspliced transcripts, and how a mature genome immune response is established, with both sense and antisense piRNAs and the methylation of histones and DNA at the provirus promoter.

## 1. Retrovirus Replication Cycle

The Retroviridae family of viruses consists of two subfamilies, *Orthoretrovirinae* and *Spumaretrovirinae*, and 11 genera [1]. The *Orthoretrovirinae* subfamily consists of the *Alpharetrovirus*, *Betaretrovirus*, *Deltaretrovirus*, *Epsilonretrovirus*, *Gammaretrovirus*, and *Lentivirus* genera. The *Bovispumavirus*, *Equispumavirus*, *Felispumavirus*, *Prosimiispumavirus*, and *Simiispumavirus* genera are in the *Spumaretrovirinae* subfamily. All retroviruses possess *gag*, *pol*, and *env* genes, although there are more complex retroviruses that encode for additional auxiliary genes such as *tat*, *tax*, *rev*, and *rex* [2]. *gag* encodes the structural proteins that form the virion, *pol* encodes the replicative enzymes, and *env* encodes glycoproteins for binding and fusion into the host cells that bear cognate receptors.

Retrovirus virions are spherical and enveloped, with a diameter of 80–120 nm [2,3]. Lining the inner surface of the viral membrane is the *gag*-encoded matrix protein (MA) [3,4,5,6]. The core of the retroviral virion is a complex fullerene lattice of a *gag*-encoded capsid protein (CA) that encapsulates replication enzymes and two copies of the viral genome [7]. HIV-1 capsid cores isolated from acutely infected cells possess all the viral components necessary to complete reverse transcription and initiate integration of the resulting cDNA into a dsDNA target [8]. The retroviral genome is single-stranded, positive-sense RNA. It generally has a 7-methyl cap structure at the 5′ end, although alternative cap structures have been reported [9,10], and it is polyadenylated at the 3′ end. The two copies of genomic RNA in the virion dimerize via hydrogen bonds and are coated with the *gag*-encoded nucleocapsid (NC) protein [11,12,13].

The virion membrane that encloses the CA core is decorated with *env*-encoded glycoproteins that recognize specific receptors on the target cell membranes to mediate adsorption and fusion, delivering the retrovirus core into the host cell cytoplasm [14]. After entry, the viral RNA is used as a template by the viral reverse transcriptase, resulting in a double-stranded cDNA that is flanked by long terminal repeats (LTRs) [15] (Figure 1). The resulting cDNA product is longer than the viral genomic RNA template because of two programmed strand-transfer steps catalyzed by the viral reverse transcriptase. The LTR is a regulatory sequence made up of the U3, R, and U5 segments [16,17]. After the integration of the viral cDNA into the host genome, the cellular RNA polymerase II (RNAP II) localizes to the U3 region of the 5′LTR and initiates the transcription of the provirus at the beginning of R. The 3′ LTR, conversely, is a transcription terminator as it contains signals for polyadenylation and RNA cleavage [17].

To generate multiple polyproteins from the single primary transcript, all retroviruses deliver an unspliced transcript and at least one spliced transcript to the cytoplasm (Figure 1). Unspliced transcripts are poorly transported into the cytoplasm; to evade nuclear retention, retroviruses have *cis*-regulatory elements. For instance, Mason–Pfizer monkey virus (MPMV) and Murine Leukemia virus (MLV) transcripts possess a cis-acting element referred to as the constitutive transport element (CTE) for export into the cytoplasm in an NXF1/NXT-dependent manner [18,19,20,21,22]. Human immunodeficiency virus type 1 (HIV-1), which makes more than 100 alternatively spliced RNAs from a single primary transcript, encodes a protein called Rev, which binds to the rev response element (RRE) of incompletely spliced HIV-1 transcripts and recruits CRM1 for nuclear export [23,24,25,26,27,28,29,30]. Transcripts are translated into viral structural and enzymatic proteins which assemble into immature viral particles, usually on the cytoplasmic face of the plasma membrane. The viral particles are then released from the plasma membrane and undergo proteolytic processing to create mature infectious viruses, which consequently infect target cells that are susceptible by virtue of expressing cognate cell surface receptors [31,32,33]. Retroviruses have obligate reverse transcriptase and integrase activities and have been exploited extensively to understand fundamental processes in molecular biology, including the regulation of gene expression [34].

## 2. Exogenous and Endogenous Retroviruses

To complete their lifecycle, retroviruses must establish a DNA provirus, which becomes a permanent genetic element within the infected cell and any daughter cells. If retroviruses are capable of transducing germline cells or the precursors of germ cells, the viruses can be transmitted vertically from parent to progeny [35,36]. Like exogenous retroviruses, such endogenous retroviruses (ERVs) could propagate by making mature viral particles which subsequently infect adjacent germline or somatic cells. ERVs alongside members of the *Pseudoviridae* (Ty1-*copia*-like), *Metaviridae* (Typ3-gypsy-like), and *Belpaoviridae* (BEL-Pao-like) families are classified as LTR-retrotransposons [1,37,38,39]. Similar to retroviruses, the genome of pseudoviruses, metaviruses, and belpaoviruses have LTRs and encode for the Gag and Gag–Pol polyproteins, with few members also expressing an *env*-like gene [40,41,42,43,44,45]. As such, they form virus-like particles and reverse-transcribe their RNA genomes using tRNA primers. However, unlike retroviruses, most members of the *Pseudoviridae*, *Metaviridae*, and *Belpaoviridae* families do not have an extracellular phase.

Since the discovery of ERVs in the late 1960s, it has been shown that ERVs constitute a large proportion of vertebrate genomes [46]. In the human genome, ERVs make up 8% of the DNA sequence and transposons more generally make up 45% [47]. However, most ERVs and other transposons do not encode functional proteins due to the accumulation of mutations. Thus, they do not have the potential to generate infectious viral particles. However, they may still transcribe RNAs, possess enhancer elements that influence host gene expression, or promote chromosome recombination. The increased expression of ERVs is associated with various cancers, autoimmune diseases, amyotrophic lateral sclerosis, and multiple sclerosis, indicating that they can be activated as part of a more generalized stress response [48,49,50,51]. These findings suggest that, despite posing a threat to the integrity of the host genome, in some cases, retrotransposition is adaptive. ERVs can even be repurposed to serve essential cellular functions. For instance, Syncytin-1 is the envelope of endogenous retrovirus-W (ERV-W) that is expressed in trophoblasts and is essential for placental development [52,53]. Retrotransposon-derived proteins can also facilitate the transfer of RNA for intercellular communication. In *Drosophila*, the activity-regulated cytoskeleton-associated protein (Arc), which is derived from retroviral Gag, modulates the transfer of its mRNA across synaptic boutons by forming extracellular vesicles [54,55]. Similarly, viral-like particles made from paternally expressed gene 10 (PEG10) mediate placenta formation by transferring RNA [56,57,58]. Additionally, ERVs are located adjacent to the regulatory regions of immune genes and the binding sites of the tumor suppressor protein p53 [59,60]. Thus, they regulate the transcriptional network of p53 and innate immunity. ERVs have also been exapted as splice donor or acceptor sites [61]. Hence, domesticated ERVs contribute to genomic evolution and are necessary for cellular functions and development.

## 3. Fixation of Endogenous Retroviruses in the Host Genome

Most ERVs are relics of ancient retroviruses that invaded the genome millions of years ago. No replication-competent endogenous retroviruses have been detected in the human genome, whereas some mouse strains have replication-competent retroviruses in their genome [62,63,64]. For example, the AKR mouse genome has two copies of the AKR gammaretrovirus that are capable of retrotransposition [65]. Currently, the Koala retrovirus (KoRV) is invading wild and captive koalas (*Phascolarctos cinereus*) in Australia [66,67,68]. KoRV is associated with lymphoma and leukemia due to the presence of viral particles in leukemic koala tissues and in mitogen-stimulated peripheral blood mononuclear cells (PBMCs) [69,70,71]. KoRV induces immunosuppression and thus increases the susceptibility of koalas to infection with chlamydia, which consequently leads to blindness and infertility, as well as lymphoid neoplasia [72,73,74]. Despite the many health consequences of infection with the KoRV retrovirus, there are currently no diagnostic assays, vaccines, or treatment regimens available.

The origins of KoRV have yet to be ascertained and remain a debatable topic. KoRV shares 78% nucleotide sequence identity to the gibbon ape leukemia virus (GaLV), demonstrating the close taxonomic relationship between the two viruses [75,76,77,78]. Gibbons and koalas, hosts of GaLV and KoRV, respectively, are evolutionarily distant and occupy different territories separated by the Wallace Line [78,79,80]. More importantly, GaLV has only been reported in captive gibbons [78]. Due to the geographical and phylogenetic distance of primates and koalas, direct transmission is unlikely [81]. Various GALV–KoRV-related viruses have been isolated from bats and rats; for instance, Hervey pteropid gammaretrovirus (HPG) and flying fox retrovirus in bats, [82,83,84], and the Melomys burtoni retrovirus (MbRV) and complete Melomys woolly monkey retrovirus (cMWMV) in rodents [85,86,87]. The phylogenetic distance between GaLV and KoRV suggests that there are yet more retroviruses that are unknown which link GaLV and KoRV [83].

KoRV is heterogeneous, with twelve different variants falling into the following three major clades based on the hypervariable region of the *env*: (a) KoRV-A, (b) KoRV-B, and (c) KoRV-C to -I and K-M [88,89,90,91,92]. KoRV-A uses the sodium-dependent phosphate transporter 1 (PiT1) receptor while KoRV-B employs the thiamine transporter 1 (THTR1) [75,92]. The other KoRV variants have low prevalence rates, and their receptors are unknown. All KoRV subtypes are spreading horizontally, but the KoRV-A subtype is also transmitted vertically from one generation to the next, with some of the literature suggesting that the germline invasion of KoRV-A began 22,200–49,900 years ago [68,77,93]. While KoRV-A proviruses are found in large numbers in the germ cells of koalas from the north of Australia—on average, each individual animal has ~70 unique integrants—there are koalas in the south that do not have KoRV-A in their germline [68,94].

Recombinants of KoRV-A, referred to as recKoRV, are observed in the germline of koalas. recKoRVs typically possess the 5′ and 3′ ends of KoRV-A and the phascolarctid endogenous retroelement (PhER) *env* and LTR in the middle [95] (Figure 2). Interestingly, even in koalas from the south that do not have endogenous KoRV-A, recKoRVs are present [95]. How recKoRVs were established in animals that do not possess KoRV-A is unclear. Nonetheless, the endogenization of KoRV-A in real time provides a unique opportunity to observe and understand how retroviral integration into germ cells leads to fixation of an endogenous retrovirus in the genome of a species, how this process affects host evolution, and how viral resistance and immunity are established.

## 4. Retrovirus Restriction Factors and Countermeasures

Host cells express restriction factors that interrupt the viral life cycle to block viral replication. Simultaneously, viruses encode proteins that antagonize these restriction factors (Figure 3). The transmembrane protein, serine incorporator 5 (SERINC5), is incorporated into HIV-1 virions and reduces infectivity by interrupting the fusion of the virus to the host cell membrane [96,97]. However, HIV-1 Nef, MLV glycoGag, and equine infectious anemia virus (EIAV) S2, counteract SERINC5 by preventing its incorporation into virions. Friend virus susceptibility gene 1 (Fv1) and tripartite motif-containing protein 5 (TRIM5) block retroviruses by interacting with the capsid immediately after entry [98,99,100,101,102]. Upon recognition of the retroviral capsids, TRIM5 forms a complementary lattice which prematurely disassembles the capsid, blocks capsid transport to the nucleus, and promotes the transcription of inflammatory cytokines via the activation of the TAK1 signaling pathway [103,104,105,106,107,108,109]. Cyclophilin A, a peptidyl-prolyl cis-trans isomerase protein, interacts with HIV-1 capsid and prevents TRIM5 from disrupting the viral capsid [110,111]. The sterile alpha-motif (SAM) and histidine-aspartate (HD) domain-containing protein 1 (SAMHD1) reduces the dNTP pool and restricts reverse transcription in non-dividing cells [112,113]. The Vpx protein encoded by HIV-2 and the related SIVs acts as an adaptor that loads SAMHD1 onto the CRL4^DCAF1^ E3 ubiquitin ligase for the subsequent proteasomal degradation of SAMHD1 [113,114]. Apolipoprotein B mRNA-editing enzyme catalytic polypeptide-like 3G (APOBEC3G), another restriction factor, introduces mutations into the reverse-transcribed DNA by deaminating cytosine into uracil [115,116,117,118]. Viral infectivity factor (Vif), an accessory protein of HIV-1, inhibits the virion incorporation of APOBEC3G [119,120,121,122]. At the end of the retroviral life cycle, tetherin inhibits the newly assembled virion particles from leaving the cell. Viral protein Vpu binds to tetherin and displaces it from the site of viral assembly [123,124]. Infection by exogenous retroviruses can thus be inhibited by preventing or interfering with the function of the viral proteins [125,126,127]. As host cells express innate immune defenses that prevent infection, viruses then express factors to overcome these defenses.

After the integration of the retrovirus into the genome, RNAP II and transcriptional factors mediate the transcription of the provirus. The host has evolved mechanisms such as Krüppel-associated box (KRAB) domain-containing zinc finger proteins (KZFPs) and the human silencing hub (HUSH) complex to restrict the retrovirus at this stage of the lifecycle (Figure 3) [128,129,130,131,132]. KZFPs are the largest family of vertebrate-specific transcriptional repressors. They recognize transposable element (TE)-embedded sequences as genomic targets and recruit transcriptional regulators such as TRIM28 (KAP1). TRIM28 consequently serves as a scaffold for the heterochromatin-inducing machinery, the heterochromatin protein 1 (HP1), the SET Domain Bifurcated 1 (SETDB1), and the NuRD complex [128,133,134,135,136]. To avoid repression by the KZFPs, transposons mutate the binding sites of the zinc finger proteins [129,137].

The HUSH complex is another host-encoded transcriptional silencing machine that is composed of the chromodomain protein MPHOSPH8 (MPP8), the RNA binding protein PPHLN1 (Periphilin 1), and FAM208A (TASOR) [138]. The importance of the HUSH complex as a barrier to retrovirus replication is emphasized by the fact that most primate immunodeficiency viruses encode accessory proteins, either Vpx or Vpr, that alleviate silencing by inducing the degradation of HUSH complex proteins [139,140]. The HUSH complex can target diverse intronless mobile elements and unintegrated retroviral DNA by recruiting MORC2, an ATP-dependent chromatin remodeler, and SETDB1 [131,132,138,141,142]. Exactly which molecular features in a retrovirus render it a target of HUSH complex-mediated silencing is a matter of ongoing investigation, but they include long coding sequences, lack of splicing/introns, and adenine-rich sequences [141].

## 5. Introduction to piRNAs

Piwi-interacting RNAs (piRNAs) are another mechanism by which host cells silence transcription of retroviruses. piRNAs were first identified as repeat-associated small interfering RNAs (rasiRNAs) in the testes of *Drosophila melanogaster* in 2001 [143]. Hyperexpression of the X-linked *Stellate* (*Ste*) genes, which encode for a protein homologous to the β subunit of protein kinase CK2, results in the formation of crystalline aggregates in spermatocytes and male sterility [144]. The *Suppressor of Stellate* (*Su(Ste)*) on the Y-chromosome is required for the production of piRNAs that repress *Ste* and maintain fertility. Moreover, the *flamenco* locus encodes piRNAs that suppress the *gypsy* family of endogenous retroviruses [145,146]. These studies demonstrated the requirement of piRNAs for gametogenesis and the suppression of transposons. The importance of piRNAs for maintaining fertility was also shown with the P element DNA transposon that invaded the *Drosophila melanogaster* genome in the 1950s. While the progeny of P strain female flies and M strain male flies are fertile, the offspring of P strain male flies crossed with M strain female flies are sterile, a phenomenon referred to as hybrid dysgenesis [147,148,149,150,151]. Maternally deposited piRNAs of P strain female flies provide protection against P elements [152,153,154]. Failure of the piRNA pathway to silence transposons causes the accumulation of double-strand breaks and the activation of the DNA damage response, which interferes with germline development [155,156]. piRNAs have since been observed in arthropods, nematodes, birds, marsupials, eutherians, and sponges [94,157,158,159,160,161,162,163].

piRNAs are loaded onto the PIWI clade of effector Argonaute proteins to form a multiprotein complex called the piRNA-induced silencing complex (piRISC). Argonautes have the following four functional domains: the N–terminal (N), PAZ (PIWI–Argonaute–Zwille), MID (middle domain), and the PIWI domains [164,165]. The MID and PAZ domains bind to the 5′ and 3′ ends of the small RNAs, respectively, and the PIWI domain possesses the RNaseH-like catalytic tetrad, DEDX (X is usually H or D), for endonucleolytic slicing [166,167,168,169,170,171,172,173]. PIWI proteins require GTSF1, a small zinc-finger protein, for their endonucleolytic activity [174]. GTFS1 binds to a piRNA-bound PIWI protein and facilitates a conformational change of the Argonaute protein to its active form. piRNAs are derived from long, single-stranded RNAs (ssRNAs). Mature piRNAs have a size range of 21–35 nucleotides, are characterized by 2′-O-methylation at the 3′ end, and a 5′-uridine or adenosine at the 10th [175,176,177,178].

## 6. piRNA Response in Fruit Flies

*Drosophila melanogaster* plays a crucial role as an animal model to study piRNAs and the piRNA pathway. In *D.melanogaster*, piRNAs are produced in somatic follicle cells and nurse cells from piRNA clusters, which are several hundreds to thousands of kilobase-long regions within the genome [179]. These piRNA clusters are enriched with fragmented and full-length transposons and are thus the genetic memory for the mobile elements that previously invaded the host germline, analogous to CRISPR (clustered regularly interspaced short palindromic repeats) arrays [179]. In the nurse cells, the HP1 homolog, Rhino, binds to the trimethylated histone 3, lysine 9 (H3K9me3) marks on the dual-strand piRNA clusters through its C-terminal chromodomain [180,181]. Such binding is stabilized by additional factors, including the zinc-finger protein Kipferl [182]. Rhino then recruits Deadlock and Cutoff to the dual-strand piRNA clusters, which lack clear signatures of RNAP II promoters [183,184,185,186,187,188]. The complex of Rhino, Deadlock, and Cutoff (RDC) initiates transcription and suppresses splicing, 5′ capping, polyadenylation, and premature transcription termination of nascent piRNA-precursor transcripts [183,187,189,190,191,192]. The Transcription/Export (TREX) complex facilitates the export and localization of these cluster transcripts to the nuage, a perinuclear electron-dense structure found near nuclear pores in the cytoplasm [193,194]. These transcripts are processed into mature piRNAs that are loaded onto the *Drosophila* PIWI proteins, Aubergine (Aub), Argonaute 3 (Ago3), and Piwi [179,195,196]. In the nuage, Ago3, loaded with an “initiator” piRNA, complementarily binds to antisense piRNA cluster transcripts thus generating a pre-pre-piRNA intermediate with a 5′ monophosphate [179,197]. This intermediate is loaded onto the Aubergine (Aub) PIWI protein and is either transferred to the outer membrane of the mitochondria via MOV10L1 (Armitage) or it remains in the nuage, where it is trimmed at the 3′ end by Nibbler (PNLCD1) as well as 2′O-methylated at the 3′ end by Hen1 (HENMT1) to enhance stability [175,198,199,200,201]. This newly generated “responder” piRNA binds to sense cluster transcripts, which are then processed to mature piRNAs loaded onto Ago3. This feed-forward amplification loop, known as the ping-pong cycle, increases piRNA abundance [179] (Figure 4). At the mitochondrial membrane, Zucchini (MITOPLD) cleaves the Aub-bound pre-pre-piRNA consecutively, from the 5′ to the 3′ direction, to generate intermediates (pre-piRNAs) with a monophosphorylated 5′ end [202,203,204,205,206]. These pre-piRNAs are primarily loaded onto Piwi, another PIWI protein, and processed into mature piRNAs [198,199,201,205,207,208,209,210,211]. This generation of trailing piRNAs from the consecutive endonucleolytic cleavage is called phasing (Figure 4) [202,203]. piRNA-bound Piwi localizes into the nucleus and recruits chromatin modifiers to establish the epigenetic silencing of transposons in a transcription-dependent manner [212,213,214,215].

In somatic follicle cells, piRNAs are derived primarily from uni-strand piRNA clusters, which have a clearly defined promoter, and like mRNAs, their transcripts are capped, spliced, and polyadenylated [179,183,216,217,218,219]. Once in the cytoplasm, Yb, a protein that contains a Tudor-domain and a DEAD-box RNA helicase domain, binds piRNA precursors and sequesters them to Yb bodies [220,221,222]. In the Yb body, the piRNA precursor is processed into a mature piRNA [222]. While all three of the PIWI proteins are expressed in the germline, only Piwi is expressed in ovarian somatic cells [179,223]. Hence, the ping-pong pathway is absent in somatic follicle cells and transposon repression occurs at the co-transcriptional level via PIWI–piRNA-induced silencing complexes (piRISCs) [212,215,219,224].

## 7. piRNA Response in Mammals

In mammals, germ cells are induced from proximal epiblasts via the bone morphogenetic protein (BMP) and WNT signaling [225,226,227]. During mammalian development, primordial germ cells (PGCs) proliferate as they migrate to the genital ridges, and are globally demethylated which consequently reactivates transposons [228,229,230,231]. Upon arrival at the genital ridge, PGCs adhere to the surrounding tissues and differentiate into either quiescent prospermatogonia or meiotic oocytes [232,233,234]. In male germ cells, the following distinct sets of piRNAs are produced: fetal pre-pachytene piRNAs, post-natal pre-pachytene piRNAs, and pachytene piRNAs [235,236,237,238,239,240,241]. Quiescent prospermatogonia produce the PIWI proteins, PIWIL2 and PIWIL4, which localize in the IMC (inter mitochondrial cement) and cytoplasmic piP bodies (P granules), respectively [236,237,242]. Due to global demethylation, the expression of both degraded and replication-competent ERVs and transposons are upregulated [229,243,244,245]. These transposon transcripts are consequently exported into the cytoplasm for processing into piRNAs.

piRNA-loaded PIWIL4 localizes into the nucleus where it binds to its cognate targets and elicits changes in gene expression through the modification of the chromatin structure [246,247,248,249,250]. Thus, prenatal piRNAs mediate the reacquisition of methylation marks and oversee the transcriptional repression of transposons. Soon after birth, PIWIL4 expression ceases, but PIWIL2 expression persists through to the round spermatid stage [236,251]. Postnatal mouse (*Mus musculus*) cells with defective PIWIL2 or PIWIL4 arrest during the early stages of meiosis I [236,237]. In golden hamsters (*Mesocricetus auratus*), however, the loss of PIWIL2 or PIWIL4 engenders a more acute phenotype [238]. Seminiferous tubules of golden hamsters deficient in PIWIL2 or PIWIL4 primarily contain sertoli cells, and the remaining germ cells arrest at the zygotene, pachytene, or diplotene stages [238] (Figure 5).

After birth, gonocytes resume mitotic proliferation and, consequently, the first round of spermatogenesis ensues [252,253]. Different sets of piRNAs mapping to repeats and the 3′UTRs of protein coding genes are expressed throughout the spermatogenic cycle [251]. However, the genomic sources of repeat-derived post-natal pre-pachytene piRNAs are different from those of prenatal piRNAs [247]. The production of the third class of piRNAs, pachytene piRNAs, begins in mice and golden hamsters at the pachytene and leptotene stages of meiosis, respectively. The appearance of pachytene piRNAs coincides with the expression of PIWIL1; the disruption of this protein results in cell cycle arrest as round spermatids or pachytene-like spermatocytes in mice and hamsters, respectively [238,254] (Figure 5). This is because the expression of pachytene piRNA genes and piRNA biogenesis factors is regulated by the transcription factors, A-MYB and TCFL5 [255,256]. Pachytene piRNA clusters originate from intergenic loci and are divergently transcribed from a bidirectional promoter [255,257]. The function of most pachytene piRNAs, however, remains unknown. Possibly, pachytene piRNAs are just selfish elements that propagate through the piRNA pathway, or they have a more passive role in the formation and maintenance of the chromatoid body (CB), which is a Ribonucleoprotein (RNP) granule. Hamsters encode an additional PIWI protein, PIWIL3, which is not expressed in the prenatal or adult testis. Hence, *PIWIL3^−^*^/*−*^ male golden hamsters are fertile, and their testes display normal morphology [238].

In females, PGCs differentiate into oogonia, which progress to the diplotene stage of prophase I and remain dormant as primary oocytes until sexual maturity [258,259,260]. During the ovarian cycle, a small number of primary oocytes complete meiosis I, forming secondary oocytes. Usually, only one secondary oocyte is released from the ovary and is drawn into the fallopian tube. After ovulation, the oocyte arrests at metaphase II (MII) until fertilization. piRNAs are dispensable for female fertility in mice and rats because they express a short isoform of Dicer (Dicer^o^) that is able to process dsRNA substrates more efficiently, and Dicer^o^-directed endogenous siRNAs effectively repress transposon mobility in mouse and rat oocytes [261,262]. Hence, golden hamsters are utilized to study the role of piRNAs in the female germline. Quiescent primary oocytes express PIWIL1, PIWIL2, and PIWIL3. PIWIL2 expression gradually diminishes, but PIWIL1 and PIWIL3 expression persists and can be detected in MII-arrested secondary oocytes and two-cell embryos [238,239,241,263]. However, piRNAs (~19) immunoprecipitate with PIWIL3 only in MII-arrested oocytes [263]. Deficiency of the PIWI proteins does not cause histological abnormalities. However, PIWIL1^−/−^ hamsters are sterile and their embryos arrest at the two-cell stage, while PIWIL3^−/−^ hamsters have reduced fecundity because fewer embryos proceed through to the later stages of embryogenesis [239] (Figure 6). This is because in PIWIL3^−/−^ oocytes, more pre-piRNAs are instead loaded onto PIWIL1 compensating for the loss of PIWIL3. Conversely, upon the loss of PIWIL1, PIWIL3 expression decreases which accounts for the more severe phenotype of maternal PIWIL1^−/−^ golden hamster embryos. Concurrent with these findings, PIWIL1 deficiency causes rogue TE expression and significantly affects the transcriptome, while PIWIL3^−/−^ does not change transposon expression [238,241]. Analogous to pachytene piRNAs in the testes, most PIWIL1- and PIWIL3-loaded piRNAs map to unannotated intergenic regions in the genome and very few of them have complementary targets. Contrary to males, however, A-MYB expression is low in the female gonad [263]. Therefore, female and male gonads do not share most piRNA precursors, suggesting the sex-specificity of piRNA clusters. Thus, overall, piRNAs modulate gametogenesis, fertility, and reproductive health by suppressing endogenous retroviruses and other transposable elements.

Consistent with the observations in animal models, pre-pachytene piRNAs dominate juvenile human testis [264]. Pre-pachytene piRNAs primarily map to protein-coding genes, and their expression remains constant through the different stages of spermatogenesis. There is abundant pachytene piRNA production in adult humans, and low expression of pachytene piRNAs is associated with azoospermia [264]. Mutations in the genes required for piRNA production, including GTSF1, PIWIL1, PIWIL2, MOV10L1, HENMT1, PLD6, and PNLCD1, impair human spermatogenesis, decrease male fertility, and are associated with the activation of transposons [265]. Although many details differ, information gleaned from animal models such as mice and hamsters have provided insight into the workings of human piRNA and PIWI systems. Due to the absence of a robust mammalian cell line that expresses the piRNA machinery, such studies require animal models making the mechanistic experiments more laborious, difficult, and time-consuming. Nonetheless, the germline spread of KoRV-A provides an exciting opportunity to study how the PIWI-mediated silencing mechanism recognizes a foreign genetic element [68,266].

## 8. Features of piRNA Precursors

There are extensive studies on piRNA defense against established ERVs and transposons. However, how the piRNA machinery initially recognizes a newly integrated, endogenous retrovirus or transposon has not been investigated as extensively. Whether transcripts that are targeted by the piRNA machinery have a unique signature that distinguishes them from other transcripts or if they are specified by preexisting piRNA complementary targets is unclear. When new retroelements invade *Drosophila*, the siRNA pathway may act as the primary repressor before the element becomes incorporated into the piRNA cluster [267,268]. How this new retroelement is first recognized as foreign by the siRNA machinery is also unclear. In koala testes, piRNAs mapping to KoRV-A have a bias towards the sense strand while piRNAs mapping to older, more established ERVs and transposons have no bias for either strand [94]. Furthermore, these piRNAs map uniformly along the length of KoRV-A, demonstrating that piRNAs are generated from the unspliced ERV transcripts. Thus, upon recognition of a newly invading, selfish genetic element, the piRNA machinery directly cleaves the transposon into piRNAs, serving as an initial, innate, genomic defense system [94]. We hypothesize that the piRNA adaptive genomic defense system is established with the production of antisense piRNAs that can specifically target the transcripts of the newly integrated selfish element (Figure 7). The preferential production of piRNAs from unspliced transcripts is observed in mice, rats, opossums, cows, and fruit flies, elucidating the deep conservation of piRNA processing from unspliced transcripts [94].

How the piRNA machinery recognizes a diverse set of ERVs and other transposon sequences, or how it differentiates other endogenous piRNA precursors from mRNAs, lncRNAs, or rRNAs, remains elusive. The observations that piRNAs are made from unspliced transcripts of endogenous retroviruses [94], that Rhino suppresses splicing of piRNA cluster transcripts in flies [190], and that some pachytene piRNA clusters have long first exons that hinder splicing [269] suggest that the lack of splicing or inefficient splicing triggers piRNA production. Thus, slow splicing or the lack thereof could act similarly to pathogen recognition receptors (PRRs) by identifying newly integrated retroviruses or endogenous piRNA cluster transcripts as non-self. Transcriptional silencing by the HUSH complex, which is thought to act independently of piRNAs or PIWI proteins, is suppressed by splicing [141]. Unspliced, intron-containing RNA produced by the HIV-1 provirus is also detected by the infected cell as a danger signal; the unspliced RNA is transported to the cytoplasm by HIV-1 Rev and the cellular protein CRM1, where it is detected by the PRR, melanoma differentiation-associated protein-5 (MDA5) [270,271,272]. Interestingly, the production of short interfering RNAs (siRNAs) from unspliced transcripts was demonstrated in the pathogenic fungus, *Cryptococcus neoformans* [273]; in this study, unspliced transposon transcripts were shown to have high spliceosome occupancy and to be inefficiently spliced due to abnormally long introns. This stalled splicing promoted the recruitment of the Spliceosome Coupled and Nuclear RNAi (SCANR) RNAi complex that was associated with the spliceosome. As in *Cryptococcus*, inefficient or absent splicing in koala germ cells could trigger piRNA production from retrovirus transcripts by mobilizing accessory proteins that facilitate piRNA production. In fact, RNAi screens of the *Drosophila* germline demonstrated that splicing facilitates piRNA production [274,275]. The accumulation of splicing factors and accessory proteins could designate a transcript as ‘non-self’ and the production of piRNAs from this transcript.

## 9. Conclusions

ERVs and other transposons are major contributors to the evolution and structure of metazoan genomes, and the control and dysregulation of their expression are critical for genomic stability, inflammation, autoimmunity, and cancer [276]. The germline silencing mechanism mediated by piRNAs and PIWI clade Argonaute proteins can discern retroviruses and other foreign elements from cellular RNA. There is no simple, common sequence motif that is conserved amongst all transposons, so it is unclear how the piRNA machinery can differentiate transposon transcripts from those of host mRNAs. Recent evidence suggests that piRNAs originate primarily from the unspliced transcripts of transposons and non-coding RNAs. In *D. melanogaster*, the Rhino–Deadlock–Cutoff (RDC) complex binds to piRNA clusters and suppresses splicing, leading to the generation of piRNAs from unspliced transcripts [190]. The generation of piRNAs from unspliced transcripts of transposons is also observed in cows, opossums, mice, and chickens, further underscoring its conservation [94]. Unspliced transcripts of KoRV-A, a retrovirus which is actively integrating into the koala genome, are selectively recognized from spliced transcripts and processed into sense-strand piRNAs [94]. Thus, the viral genomic RNA is directly degraded. This suppression of viral replication serves as an innate defense mechanism until antisense piRNAs are produced and adaptive immunity is established (Figure 7). Hence, the absence of splicing or inefficient splicing may be the basis for recognition by an innate genome immune system. However, no study has been conducted to determine the role of splicing in piRNA processing. Questions such as “What is the spliceosome occupancy of piRNA precursors?”, “Does the splicing machinery recruit piRNA accessory proteins?”, and “How does the piRNA machinery recognize a novel genomic invader?” remain unanswered. It is also quite interesting how silencing by disparate genomic defense systems such as the HUSH complex and piRNAs can be circumvented by splicing and/or the presence of introns.

## Figures and Tables

**Figure 1 viruses-16-01756-f001:**
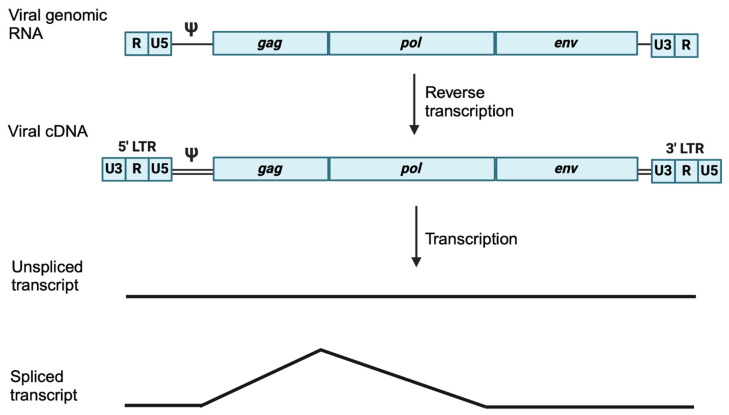
Retroviral genomic RNA and its transformations. Shown are schematic diagrams for the virion-associated genomic RNA, the viral cDNA, and the unspliced and spliced transcripts that are common to all retroviruses. All retroviruses possess at least the three genes, *gag*, *pol*, and *env*. Note that during reverse transcription, two sequential strand-exchange reactions extend the 5’ and 3’ ends of the cDNA beyond the limits of the genomic RNA template.

**Figure 2 viruses-16-01756-f002:**
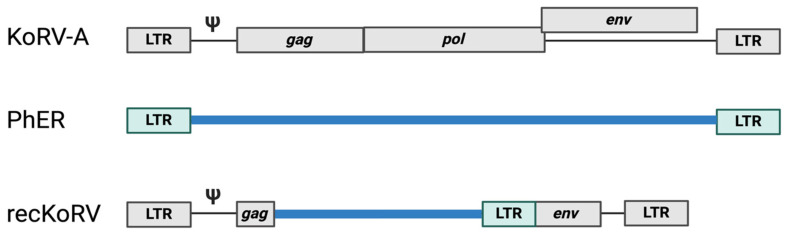
Structure of recKoRV. The koala retrovirus, KoRV-A (shown in gray), encodes gag, pol, and env with long terminal repeats at the ends. PhER (shown in blue), is an endogenous retrovirus with no protein coding capacity. Recombinant KoRV (recKoRV) typically contains the KoRV-A 5’ LTR, truncated gag, truncated env, and 3’ LTR with the 3’end of PhER in the middle.

**Figure 3 viruses-16-01756-f003:**
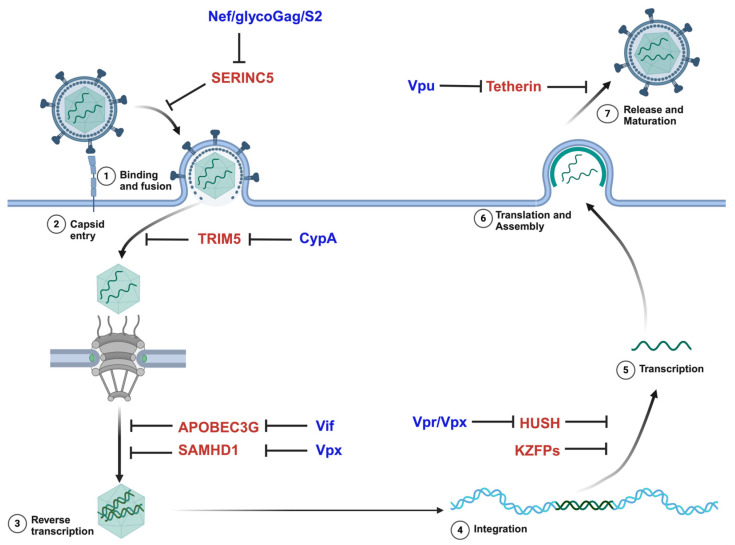
Host restriction factors and retroviral antagonists. Restriction factors are shown in red and viral antagonists are shown in blue. CypA: cyclophilin A; KZFPs: Kruppel-associated box (KRAB)-containing zinc finger proteins; HUSH: human silencing hub (HUSH) complex; Vpr: Viral protein R: Vif: Viral infectivity factor; Vpu: Viral protein U; APOBEC3G (apolipoprotein B mRNA editing enzyme, catalytic subunit 3G); SAMHD1: SAM domain and HD domain-containing protein 1.

**Figure 4 viruses-16-01756-f004:**
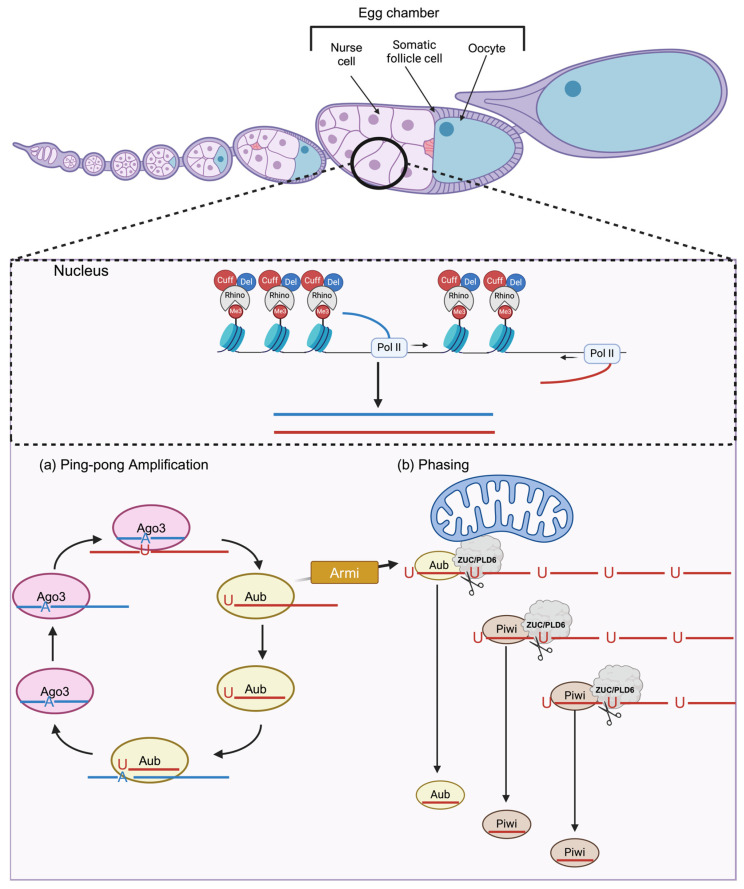
piRNA biogenesis in nurse cells of Drosophila ovaries. *D. melanogaster* ovaries contain a series of developing egg chambers in linearly arranged repetitive strings called ovarioles. An egg chamber is characterized by a germline cyst, which contains 15 germline nurse cells and an oocyte that is surrounded by somatic follicle cells. In the nurse cells, germline dual-strand clusters decorated with H3K9me3 marks bound by Rhino-Deadlock-Cutoff (RDC) complex are transcribed by RNA Polymerase II. These transcripts are exported into the cytoplasm, where they are processed into mature piRNAs by the ping-pong amplification loop or phasing. (**a**) Ping-pong amplification: The feed forward cleavage of complementary transcripts by Aub and Ago3 results in piRNAs with a 10-nucleotide overlap. (**b**) Phasing: Armi shuttles Aub bound to a piRNA precursor to the mitochondria where Zucchini generates piRNA intermediates through cleavage adjacent to uridines along the length of the precursor. These piRNA intermediates loaded on Piwi are then processed into mature piRNAs.

**Figure 5 viruses-16-01756-f005:**
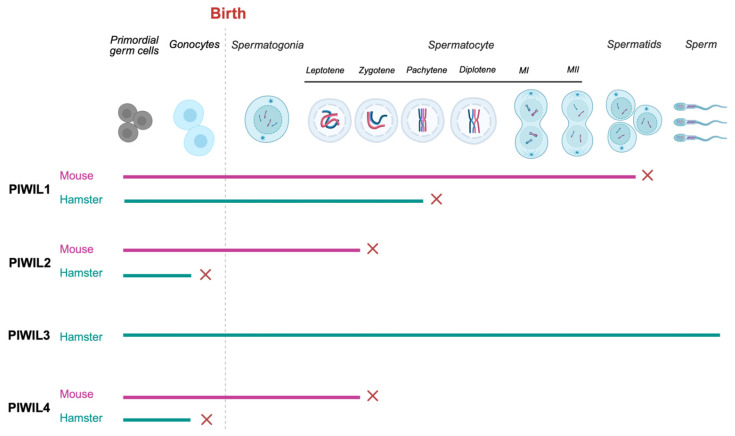
Spermatogenic defects of PIWI mutants in mice and hamsters. In mice, PIWIL2 and PIWIL4 mutants arrest at the zygotene stage of meiosis I and PIWIL1 mutants arrest at the round spermatid stage. In hamsters, PIWIL3-KO does not cause any defect in the testes. PIWIL1-KO results in arrest at the pachytene stage. PIWIL2 and PIWIL4 defective hamsters arrest during mitosis as gonocytes. Solid lines show normal development; red crosses (x) indicate the stage of developmental block.

**Figure 6 viruses-16-01756-f006:**
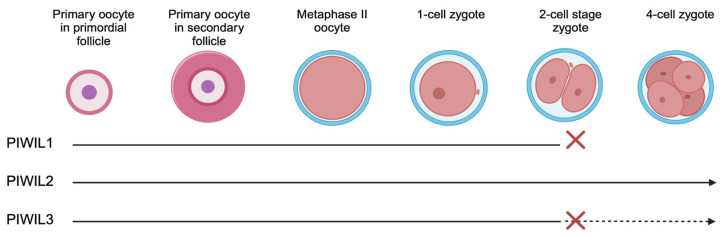
Oogenic defects of PIWI mutants in hamsters. PIWIL1 deficiency results in arrest at the 2-cell stage. PIWIL2-KO mutants have no defects in oocytes. PIWIL3 deficient hamsters arrest at the 2-cell stage, but some fertilized oocytes complete development. Solid lines show normal development; red crosses (x) indicate the stage of developmental block.

**Figure 7 viruses-16-01756-f007:**
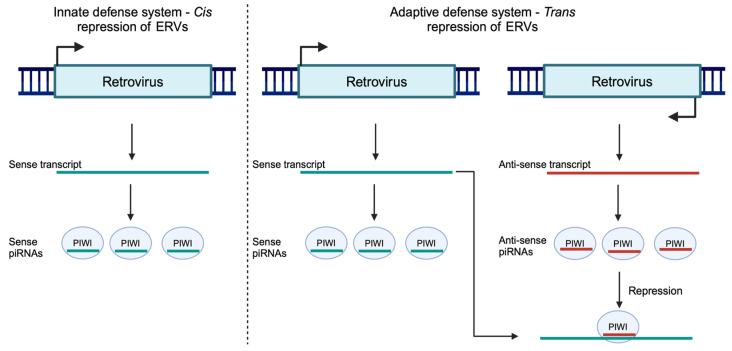
Model of innate and adaptive piRNA genome defense. Upon invasion of the germline by a novel retrovirus, the retroviral transcript is directly processed into positive sense piRNAs. Later, the adaptive piRNA response is established where antisense piRNAs are made. These antisense piRNAs can directly target the sense transcript resulting in the co-transcriptional repression of the transposon.

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
