# Peer review of "piRNA Defense Against Endogenous Retroviruses"

_viruses, 2024, doi:10.3390/v16111756_

Round 1
Reviewer 1 Report
Comments and Suggestions for Authors
The authors have provided a nice overview of retrovirus/erv restricting mechanisms, I have learnt a lot by reading it.
Title on piRNAs is maybe a bit too specific.
What is known about piRNA mediated control of erv/HERV in pathological conditions? In light of the emerging importance of activated HERV elements in a variety of diseases could the authors elaborate on whether and how such piRNA mediated control could be translated into a possible therapeutic approach?
piRNA modulation also applicable when no virions are generated so as in HERV-W vs HERV-K?
Retroviral damage is not only due to DNA damage/mutation but also unwanted transcription of viral genes or modulation of neighbouring genes
Line 121: please explain better: “have incorporated innate immune binding sites within their genomes and, in the process of retrotransposition, distributed the binding sites throughout host genomes [45]. have incorporated innate immune binding sites within their genomes and, in the process of retrotransposition, distributed the binding sites throughout host genomes [45]."
Fig. 3 needs better caption and explanation
KoRV-A invasion is of interest but the author somehow missed to report on current observations and conclusions? Maybe this was also lost in the very detailed mechanical discussion of involved RNA species and proteins?
Reviewer 2 Report
Comments and Suggestions for Authors
The review of Abajorga and co-authors focuses on the role of the piRNA pathway, emphasizing its function in protecting against endogenous retroviruses. The authors thoroughly describe the mechanisms of piRNA biogenesis and function in both Drosophila and mammals. Special attention is given to the interplay between splicing and piRNA production. Based primarily on studies of the koala retrovirus, the authors propose a model how the invasion of a new ERV may trigger a piRNA response. The central idea—that retroviruses’ tendency to produce unspliced transcripts may serve as a marker of foreignness—is intriguing, though the underlying mechanisms of this phenomenon remain unclear.
While the review is overall a nicely written piece, its structure raises some concerns. Additionally, certain significant sections lack references, and some discussions appear inconsistent.
Major Comments:
Structurally, the manuscript reads more like two separate reviews: one on endogenous retroviruses and the other on the piRNA pathway. The section on the piRNA pathway covers many aspects that do not directly relate to the main theme of the review. To improve coherence and readability, the authors could consider trimming down some of the extraneous details in piRNA section that are unrelated to the core ideas presented.
At the same time, it would make sense for the piRNA section to focus more on the various mechanisms involved in the endogenization of retroelements. For instance, studies have shown that when new retroelements invade Drosophila, the siRNA pathway may act as the primary repressor before the element becomes incorporated into the piRNA cluster (see https://pubmed.ncbi.nlm.nih.gov/23079419/, https://pubmed.ncbi.nlm.nih.gov/20581131/, and other related works).
Moreover, given that a substantial portion of piRNA studies has been conducted on retrotransposons, it would be beneficial to discuss the potential evolutionary relationship between ERVs and LTR-retrotransposons within the manuscript.
Specific Comments:
Line 26: "piRNAs are processed from unspliced transcripts of piRNA clusters, genomic loci from which most piRNAs originate." Since flamenco transcripts are spliced, along with other examples of spliced piRNA clusters, it would be more accurate to mitigate this statement by adding “mostly or preferentially.”
Line 90: " In the cytoplasm, viral structural and enzymatic proteins are made from the transcripts to assemble a retroviral particle " This should be rephrased as: "Transcripts are translated into viral structural and enzymatic proteins ..."
Lack of References: Some important statements are missing references. Specifically, lines 464-471, which describe the relationship between splicing and RNAi/piRNA, as well as lines 254-259 and 261-267.
Line 277: "How Rhino distinguishes H3K9me3 marks deposited on piRNA clusters from bulk heterochromatin is unclear." At least one study offers an explanation for why Rhino is not recruited to bulk heterochromatin, such as satellite repeats (see https://pubmed.ncbi.nlm.nih.gov/36193674/).
Lines 321-323: "Hence, the ping pong pathway is absent in somatic follicle cells, and transposon repression occurs at the co-transcriptional level via Piwi-piRNA induced silencing complexes (piRISCs) [154,180–184]." References 180-184 do not directly support this statement. It would be more appropriate to cite works that specifically demonstrate Piwi role in the transcriptional silencing of transposable elements.
Figure 3: The figure shows clearly defined promoters on the borders of piRNA clusters, which is not accurate for germline dual-strand piRNA clusters.
Line 351: "Soon after birth, PIWIL4 expression ceases, but PIWIL2’s expression persists through to the round spermatid stage (Fig. 4)." The reference to Figure 4 in this sentence is somewhat confusing, as the figure illustrates developmental arrests in piwi mutants rather than the timing of Piwi expression. Also, change "PIWIL2’s expression" to "PIWIL2 expression."
Lines 471-472: The statement, "In fact, RNAi screens of the Drosophila germline demonstrate that splicing facilitates piRNA production [222,223]," somewhat contradicts earlier sentences, which imply that the absence of splicing is a hallmark of piRNA production. Additional clarification is needed here to explain what the authors mean regarding the role of splicing in piRNA production.
Comments on the Quality of English Language-
Reviewer 3 Report
Comments and Suggestions for Authors
In the review titled “piRNA defense against endogenous retroviruses,” authors provide an impressively comprehensive review of literature about how endogenous retroviruses interact with animal genomes and how these genomes defend themselves using the PIWI-interacting RNA (piRNA) pathway. The review article is very clearly written, with nearly every sentence loaded with useful information, and contains a very comprehensive representation of the publications from across the piRNA field. In the abstract of the article, the authors offer a hypothesis (Line 38), which seems a bit out of place considering the aim of the article is a review of the field, and there is no possibility to address any hypothesis. If authors want to include this sentence, I suggest doing so in the appropriate section of the abstract.
I found the paragraph on lines 241-259 written slightly confusing. The early sentences lump all the small RNA pathways together, making it appear they come to be configured similarly. However, while the RNAs’ protein partner at the core is from the same family, the biogenesis pathways that lead to final complexes vastly differ between piRNA and miRNAs, for example. Although the authors briefly mention the biogenesis proteins later on (Dicer and Drosha), the biogenesis distinction is still not very obvious. I would suggest including a few sentences in this regard.
In the paragraph on lines 284-305, nearly every line starts with this or these. I suggest the authors adjust this so that this paragraph does not sound overly repetitive.
Best regards,
